# Early Glottic Cancer Treated by Transoral Laser Surgery Using Toluidine Blue for the Definition of the Surgical Margins: A Pilot Study

**DOI:** 10.3390/medicina56070334

**Published:** 2020-07-03

**Authors:** Eugenia Allegra, Maria Rita Bianco, Chiara Mignogna, Gaetano Davide Drago, Domenico Michele Modica, Lidia Puzzo

**Affiliations:** 1Otolaryngology, Department of Health Science, University Magna Graecia of Catanzaro, Viale Europa, Germaneto, 88100 Catanzaro, Italy; mrbianco@unicz.it (M.R.B.); g.davide.drago@gmail.com (G.D.D.); domenicomichelemodica@gmail.com (D.M.M.); 2Anatomic Pathology Unit, Pugliese-Ciaccio Hospital General Hospital, Interdipartimental Sevice Center, University Magna Graecia of Catanzaro, 88100 Catanzaro, Italy; mignogna@unicz.it; 3Department G.F. Ingrassia, Section of Anatomic Pathology, University of Catania, 95100 Catania, Italy; lipuzzo@unict.it

**Keywords:** early glottic cancer, laryngeal cancer, transoral laser microsurgery, cordectomy, toluidine blue, resection margins, head and neck cancer

## Abstract

*Background and objectives*: Transoral laser microsurgery (TLM) is widely accepted for its advantages, which consist of a brief hospital stay, rapid functional recovery, low management costs and the fact that it can be easily repeated in cases of recurrence. However, a high incidence of positive or narrow surgical margins has been reported in the literature, even if controversy still exists on the prognostic significance of positive resection margins. The aim of the study was to evaluate the utility of toluidine blue staining in defining the resection margins of early glottic cancer (T1a–T2) treated with TLM. *Materials and Methods*: This retrospective study was conducted on patients with early glottic cancer (T1a–T2) managed by TLM. A group of patients treated between 2010 and 2014 underwent toluidine blue staining (TB group) of the lesions before starting the cordectomy by TLM, and a group of patients treated by TLM between 2006 and 2009 was considered the control group. *Results*: A total of 44 subjects were included in this study: 41 were men, and 3 were women. The mean age was 58 ± 9.0 years (median 59.0, range 41–77). Twenty-three of the 44 patients were included in the TB group and 21 in the case control group. In the TB group, only the positivity of the deep margin was a predictor of local recurrence (*p* = 0.037), while in the control group, positive or close margins and the type of cordectomy were predictive factors of local recurrence (*p =* 0.049). Considering the TB group and control cases, the 5-year local recurrence-free survival was 95.6% and 80.9%, respectively (*p =* 0.14). *Conclusions*: From this first study, toluidine blue staining seems to be a useful modality to improve the rate of the negative resection margins of early glottic cancer (T1a–T2) treated by TLM.

## 1. Introduction

Laryngeal squamous cell carcinoma represents 4.5% of all malignancies and is the most common type of head and neck cancer in males [1]. Tumor biology, clinical behavior and prognosis differ on the basis of its location: glottis, supraglottis or subglottis region [2]. Consequently, the methods of treatment vary according to the location, as well as the stage of the disease [3], from organ preservation strategies such as radiotherapy [4], transoral laser microsurgery (TLM) [5,6], and open horizontal partial laryngectomy [7] to demolitive surgery (total laryngectomy). The diagnosis of glottic cancer, more often than other laryngeal sites, is made in the early stage (Tis–T1–T2) due to the early onset of clinical symptoms, such as hoarseness or progressive dysphonia. Early glottic cancer can then benefit from different conservative modalities of treatment, including radiotherapy (RT) [8], transoral laser surgery [9,10,11] or open horizontal partial laryngectomy (OHPL) [7,12], especially in cases of anterior commissure involvement, with good and similar functional and oncological results [13]. However, for the T1–T2 middle third of the vocal cords, the two main treatment modalities are RT and TLM, and although there is evidence that TLM is associated with a higher laryngeal preservation rate than RT [8,14,15], the treatment strategy depends on the preference of the surgeon and patient. TLM is widely accepted for its advantages, which consist of a brief hospital stay, fast functional recovery, low management costs [9] and the fact that it can be easily repeated in cases of recurrence. However, a high incidence [16,17] of positive or narrow surgical margins has been reported in the literature, even if controversy still exists on the prognostic significance of positive resection margins. The finding of narrow or positive margins of surgical resection constitutes a condition that poses serious questions to the surgeon on how to proceed: re-intervene or take a wait-and-see attitude? For this reason, it would be desirable to have techniques that facilitate the visualization and detection of lesions to perform an exeresis with negative resection margins on the first treatment. In the last decade, toluidine blue staining has been used mainly for the detection of the oral cavity, oropharynx and esophagus premalignant and malignant lesions [18,19,20,21]. Toluidine blue is an acidophilic metachromatic dye that selectively stains acidic tissue components. In vivo, toluidine blue stains deoxyribonucleic and nucleic acids and may be retained in intracellular spaces of dysplastic epithelium [22]. The test is based on the fact that dysplastic cells may contain quantitatively more nucleic acids, which facilitates the retention of the dye in precancerous and cancerous cells, which are replicating in vivo, whereas normal mucosa fails to retain the dye [23]. The aim of the study was to evaluate the utility of toluidine blue staining in defining the resection margins of early glottic cancer (T1a–T2) treated with TLM.

## 2. Materials and Methods

This retrospective study was conducted on patients with early glottic cancer (T1a–T2) managed by transoral laser microsurgery (TLM). The inclusion criteria were as follows: patients with small and superficial early glottic cancer previously submitted to biopsy. The exclusion criteria were the involvement of the anterior commissure, previous treatment by surgery or radiotherapy, and loss to follow-up. Twenty-three patients treated between 2010 and 2014 underwent toluidine blue staining (TB group) of the lesions before starting the cordectomy, while 21 patients treated between 2006 and 2009 were considered as a control group. All patients were treated by the same surgeon (A.G.). The study was approved by the Institutional Review Board of the University of Catanzaro. All patients were informed of the benefits, risks, possible complications, alternatives to surgery and use of toluidine blue (TB group) before giving informed consent for the surgery. All patients preoperatively underwent videolaryngoscopy with flexible endoscopy and videolaryngostroboscopy. A preoperative CT (Computed Tomography) or MRI (magnetic resonance imaging) scan was performed in both groups to confirm tumor extension [24], except for small superficial lesions limited to the middle third of the vocal cords. All patients underwent TLM under general anesthesia using a Co2 Laser (GA-0000560B N° 284, Lumenis, Yokneam, Israel) mounted on a Zeiss surgical microscope that was provided by a video camera with a recording system. The type of cordectomy performed was classified according to the European Laryngological Society [7] and was performed by the senior surgeon, A.G. All surgical specimens were three-dimensionally oriented, using suture threads with different fine needles, for the anterior and posterior superficial edges to be evaluated by the pathologist. Whenever possible, the lesions were removed en bloc, and in bulky tumors, a multibloc procedure was utilized. In all cases, the surgeon tried to achieve a margin of healthy tissue of at least 1 mm. The original primary tumor slides were retrospectively reviewed by two experienced pathologists (L.P. and C.M.). Resection margins were classified as free if ≥1 mm, close if <1 mm free, and positive if infiltrated by tumor cells. All patients were followed up 1 month after surgery, every 3 months for 3 years, and every 6 months thereafter. Local recurrence was defined as the occurrence of a new lesion less than 2 cm from the primary site, while local recurrence free survival was calculated as the time from surgery to the first local recurrence. Demographic and clinical patient data were retrieved from a clinical data collection database.

### 2.1. Toluidine Blue Staining

Toluidine blue staining of lesions was performed before starting cordectomy. A small cotton pad attached to a micro caliper valve was used to proceed with the staining method. The mucosa of the glottic plane, ventricle and false cords were cleaned twice with sterile water for 20 s, and then a solution of 1% acetic acid was applied for 20 s, followed by a 1% toluidine blue stain application for 20 s. Again, the mucosa was treated twice with the 1% acetic acid solution to reduce the extent of the mechanically retained stain. Finally, the mucosa was again cleaned with sterile water (Figure 1). The interpretation is based on the color: a dark blue (royal or navy) stain is considered positive, light blue staining is doubtful, and when no color is observed, it is interpreted as a negative stain, as previously described [18].

### 2.2. Statistical Analysis

The statistical analysis was performed with MedCalc software Version 19.4 (Mariakerke, Belgium) using the chi-square test and Fisher’s exact test. Correlations between groups and the clinical data were examined with the Mann–Whitney U test. A multivariate analysis was performed using multiple regression to determine the independent prognostic factors. The Kaplan–Meier method was used to evaluate the 5-year local recurrence and disease-free survival, and the log-rank test was used to compare survival curves between groups. A *p*-value less than 0.05 was considered statistically significant.

## 3. Results

A total of 44 subjects were included in this study: 41 were men, and 3 were women. The mean age was 58 ± 9.0 years (median 59.0, range 41–77). The mean ± SD follow-up time was 93.3 ± 30.4 months. According to the 7th edition of the TNM classification established by the American Joint Committee for Cancer, 18 lesions (40.9%) were staged as T1a N0 and 26 (59.1%) as T2N0. Of the 44 patients, five were submitted to Type I cordectomy, six type II, eighteen type III, ten type IV and five type V. On the histological examination, 30 of the 44 (68.1%) patients had disease-free margins, while 14 (31.9%) had at least one positive or close margin. Of the four patients with positive margins, three underwent re-intervention by TLM and one underwent supracricoid laryngectomy because of anterior commissure involvement. These patients were not included among those with recurrence after the first treatment as they underwent a revision of the first intervention for persistence of disease (positive resection margins). In the remaining 10 patients, we applied a “wait-and-see” approach with a strict follow-up every month for the first 6 months after the operation. During the follow-up period, five of the 44 patients (11.3%) presented with local recurrence after a mean time of 16.8 months (range: 7–30 months). Two of them had T1a, and three had a T2 tumor. Two of them were treated by radiation therapy, two by supracricoid laryngectomy and one by salvage total laryngectomy and neck dissection followed by radiation therapy. The 5-year local recurrence-free survival rates were 96.6% and 71.4% for patients with negative and positive/close margins, respectively, *p =* 0.01 (Figure 2). Twenty-three of the 44 patients were included in the TB group and 21 in the case control group. There were no significant differences in the demographic and clinical data between the two groups (Table 1). Regarding the resection margins, 4 of the 23 (17.3%) patients in the TB group had positive/close margins, while 10 of the 21 (47.6%) patients in the control group had positive/close margins. By analyzing the number of patients who did or did not develop a recurrence, in the TB group 1/4 (25%) of patients with positive close/positive margins developed a relapse while none of the negative patients developed a relapse. The correlation between the TB group and the control group (Table 2) revealed that the TB group showed less positive/close margins, especially for superficial margins (*p =* 0.047). We stratified the data according to the tumors that were negative margins and those that were positive/close margins (Table 3). Univariate analysis revealed that patients had significantly more negative margins in the TB group than in the control group (82.6% versus 53.3%; *p =* 0.01), and patients with negative margins had less local recurrence (*p =* 0.029). A multivariate analysis indicated that positive resection margins were the only independent negative predictive factor of local recurrence (0.001). In the TB group, only the positivity of the deep margin was a negative predictor of recurrence (*p* = 0.037), while in the control group, positive/close margins and the type of cordectomy were predictive factors of local recurrence (*p =* 0.049). Considering the TB group and control cases, the 5-year local recurrence-free survival was 95.6% and 80.9%, respectively (*p =* 0.14) (Figure 3), this difference was not statistically significant.

## 4. Discussion

In the treatment of T1–T2 early glottic tumors with TLM, the main risk factors are the involvement of the anterior commissure and adequate laryngoscopic exposure of the glottic plane, especially from the status of the resection margins [11,16]. These conditions predispose the patient to local recurrence and require further treatment to obtain oncological radicality [6,25]. Recently, some studies have proposed new modalities for the treatment of patients with glottic plane exposure difficulties [26,27], while other studies are aimed at finding intraoperative markers that can better demarcate the lesion to be removed. Piazza et al. [28] studied the effect of intraoperative narrow banding imaging (NBI) during transoral laser microsurgery, a novel endoscopic technique using filtered wavelengths to detect the microvascular abnormalities associated with preneoplastic and neoplastic lesions. The use of NBI during transoral laser microsurgery for early glottic cancer shows that NBI reduces the incidence of positive superficial surgical margins [29]. However, this technique is expensive and requires a learning curve—both factors that play very important roles. The application of toluidine blue for the detection of premalignant and malignant lesions of the oral cavity was reported for the first time in the 1960s [30]. These authors showed that toluidine blue was able to detect premalignant and malignant lesions of the oral cavity mucosa that were previously determined to be clinically suspicious. Toluidine blue staining is used on the basis of its ability to bind to acidic tissue components, such as the nuclear material of tissues with high DNA, and RNA contents, such as dysplastic and neoplastic cells that contain quantitatively more nucleic acids than normal tissues. Many other studies have been performed since that time to determine the sensitivity and specificity of in vivo TB staining. In 1989, a meta-analysis [31] assessing the effectiveness of toluidine blue application in the identification of oral squamous cell carcinoma determined the sensitivity to be in the range of 93.5% to 97.8%, and specificity to be in the range of 73.3% to 92.9%. According to more recent studies, sensitivity ranges from 86% to 100%, and specificity ranges from 44% to 100% [32,33,34]. Reviewing the literature, we found that Portugal et al. [35] used toluidine blue for the assessment of the intraoperative tumor margin status after the resection of oral and oropharyngeal primary tumors. They found that toluidine blue staining improved the ability to assess the intraoperative margin status at the time of resection with a sensitivity of 100% and a specificity of 97%. The use of TB staining in carcinomas of the larynx has been reported up to now only by Kurniawan et al. [36]. In their case reports, TB staining had been able to identify positive and negative resection margins out of five cases of laryngeal carcinoma. In our study, toluidine blue staining was used during the transoral laser microsurgery to assess the intraoperative margin status in early glottic cancer to improve the rate of negative resection margins. The study has shown that TB is reliable in the evaluation of the superficial margins (*p* = 0.047), while it has not proven useful in the deep margin assessment for its poor power to penetrate the deep layers of the mucosa. However, the results obtained seem to be encouraging; in fact, we found a significant difference in the number of patients with negative resection margins between the TB group and the control group (82.7% versus 52.4%, respectively; *p =* 0.047) and consequently the 5-year local recurrence-free survival rates were 95.6% and 80.9% in the TB group and control group, respectively. This difference does not reach statistical significance, probably due to the limited number of cases.

### Limitations of the Study

This study has some limitations represented by the small number of patients and the comparison of data with a group consisting of historical control patients; therefore, it represents a pilot study. We believe that the use of intraoperative toluidine blue staining has some advantages since it is inexpensive, it does not require a highly expensive methodology and tools, it is simple and rapid, and it is a less technique-sensitive method.

## 5. Conclusions

From this first study, we feel that toluidine blue staining can be a useful modality to improve the rate of negative resection margins of early glottic cancer (T1a–T2) treated by TLM. However, future research will need to be performed to strengthen the role of TB, such as with a larger cohort, so that such implications can be made.

## Figures and Tables

**Figure 1 medicina-56-00334-f001:**
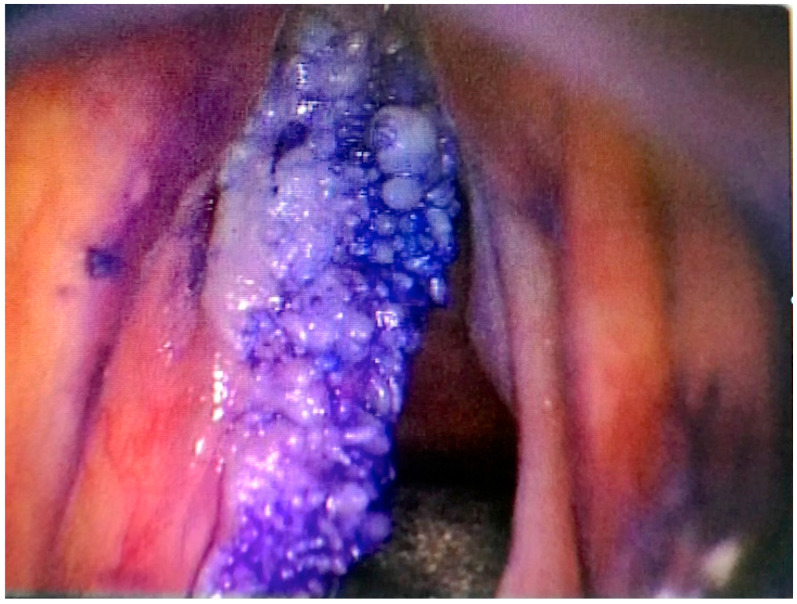
Aspect of a T2 glottic cancer after toluidine blue staining.

**Figure 2 medicina-56-00334-f002:**
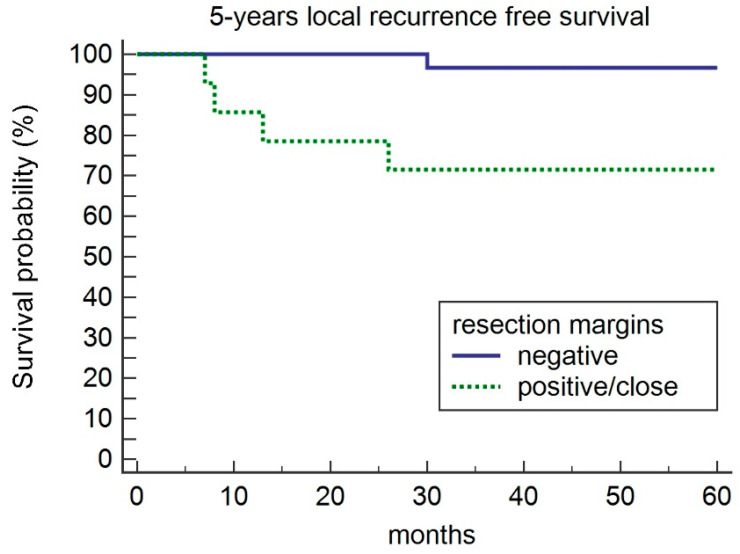
Five year local recurrence-free survival according to resection margins.

**Figure 3 medicina-56-00334-f003:**
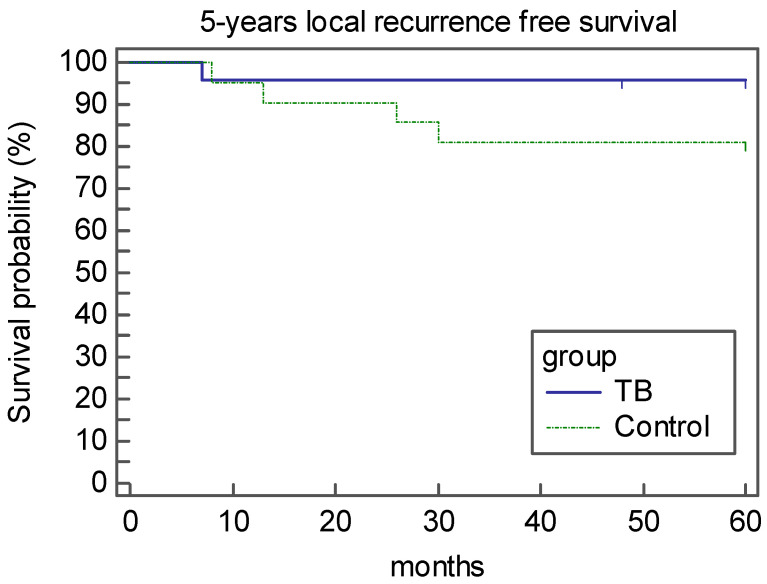
Five year local recurrence-free survival between the TB group and the control group.

**Table 1 medicina-56-00334-t001:** Analysis of the clinical and demographic data between the toluidine blue staining (TB) and control groups.

Clinical Data	Group TB n. 23	Control Group n. 21	*p*
**Age**			
Mean	58 ± 8.9	57 ± 9.4	
(range)	(44–77)	(41–77)	0.71
**Sex**			
Female	2	1	
Male	21	20	1.0
**Smoking**			
Yes	21	18	
No	2	3	0.65
**Cordectomy**			
I	3	2	
II	3	3	
III	10	8	
IV	4	6	
V	3	2	0.9
**T stage**			
I	9	9	
II	14	12	1.0

**Table 2 medicina-56-00334-t002:** Pathological status of the superficial and deep resection margins in the TB and control groups.

Group	Superficial Margins		Deep Margins	
	close/+	−	close/+	−
TB	3	20	2	21
Control	9	13	7	14
*p*	0.047		0.06	

**Table 3 medicina-56-00334-t003:** Analysis of the clinical and demographic data according to resection margins status.

Clinical/Demographic Data	Negative Margins n. 30	Close or Positive Margins n. 14	*p*
**Age**			
Mean (SD)	58.7 (8.7)	56.3 (9.8)	0.41
**Sex**			
F	1	2	
M	29	12	0.23
**Smoker**			
Yes	27	12	
No	3	2	0.64
**T Stage**			
I	14	4	
II	16	10	0.33
**Cordectomy**			
I	4	4	
II	11	1	
III	11	1	
IV	3	4	
V	1	4	0.53
**Group**			
TB	19	4	
Control	11	10	0.01
**Recurrence**			
Yes	1	4	
No	29	10	0.029

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
