# Peer review of "Early Glottic Cancer Treated by Transoral Laser Surgery Using Toluidine Blue for the Definition of the Surgical Margins: A Pilot Study"

_medicina, 2020, doi:10.3390/medicina56070334_

Round 1
Reviewer 1 Report
This is a well written and interesting paper piloting the use of in vivo application of toluidine blue to assess margins in laryngeal cancer. The study methodology is acceptable, and as the author's note, a good starting point for assessment in a more formal trial.
Due to the methodology there are several questions that came up as I read the manuscript, that the authors should address.
1) Were all cases done by the same surgeon? There were a relatively low volume of cases (4-5 cases per year on average). What was the level of experience at the institution with these cases in 2009. Was it well established or just starting out? This may be another explanation for the decrease in margin rate between groups.
2) Similarly, assessment of the deep margin is not affected by toluidine blue. The difference between groups did not reach clinical significance, but the difference in positive margins between the control and the toluene blue is of the same magnitude as the difference in mucosal margin positivity. Do the author's have a potential explanation for this observation? My question reading the manuscript was if it was due to increasing experience.
3) Toluidine blue has been reported for laryngeal cancer before, and while this level of analysis has not been done the authors should cite pre-exisiting work including:
Kurniawan AN, Handikin LS. Some experience with in vivo staining of early laryngeal cancer in Jakarta, Indonesia. CancerKurniawan AN, Handikin LS. Some experience with in vivo staining of early laryngeal cancer in Jakarta, Indonesia. Cancer Detect Prev. 1981;4(1-4):313‐317.
Detect Prev. 1981;4(1-4):313‐317.
One additional comment is regarding the author's reported downsides of NBI. NBI is often a potential feature of many endoscopic towers, and the additional cost is often quite negligible (unless otherwise serviceable pre-existing equipment had to be replaced.)
Author Response
ANSWERS TO THE REVIEWER 1
1) Were all cases done by the same surgeon? There were a relatively low volume of cases (4-5 cases per year on average). What was the level of experience at the institution with these cases in 2009. Was it well established or just starting out? This may be another explanation for the decrease in margin rate between groups.
All patients were operated on by the same surgeon (Prof. Aldo Garozzo), who at the time had about 40 years of experience and was Director of the ENT Unit. The number of patients recruited in the study does not represent the total number of T1-T2 laryngeal carcinoma patients treated per year in our Institution.
2) Similarly, assessment of the deep margin is not affected by toluidine blue. The difference between groups did not reach clinical significance, but the difference in positive margins between the control and the toluene blue is of the same magnitude as the difference in mucosal margin positivity. Do the author's have a potential explanation for this observation? My question reading the manuscript was if it was due to increasing experience.
The TB ability to penetrate into the deep layers of the mucosa is not known and the study has shown that TB is reliable in the evaluation of the superficial margins (P = 0.047), while not proven useful in the deep margin assessment (Lines 148-149 and table 2).
3) Toluidine blue has been reported for laryngeal cancer before, and while this level of analysis has not been done the authors should cite pre-exisiting work including:
Kurniawan AN, Handikin LS. Some experience with in vivo staining of early laryngeal cancer in Jakarta, Indonesia. CancerKurniawan AN, Handikin LS. Some experience with in vivo staining of early laryngeal cancer in Jakarta, Indonesia. Cancer Detect Prev. 1981;4(1-4):313‐317.
We thank the Reviewer for the suggestion certainly useful to improve the text. We added the citation and comment it in the discussion (Lines 234-235)
4) One additional comment is regarding the author's reported downsides of NBI. NBI is often a potential feature of many endoscopic towers, and the additional cost is often quite negligible (unless otherwise serviceable pre-existing equipment had to be replaced.)in the text.
We agree with the Auditor. The NBI does not entail an excessive additional expense in the case of a new purchase of an endoscopic tower while for those who already have an endoscopic tower they must bear a new expense to have the NBI. Furthermore, as specified in the text, TB is almost free of charge and does not require a learning curve.
Reviewer 2 Report
The authors describe a well known technique using Toluidine Blue for a better demarcation of small laryngeal cancers during transoral laser surgery.
Major remarks:
- 75: in cases of T1 laryngeal carcinomas: could previous biopsies influence TB staining?
- 77-78: 23 TB and 21 control patients in 3-4 years i.e. six/seven T1-2 laryngeal cancer patients each year on average: are these all T1-2 laryngeal cancer patients treated in your institute during this period? Are they consecutive patients or has there been a selection (bias)? This should be explained.
- 124-125: Is it statistically sound to evaluate recurrence free survival after a second intervention ? If these patients are excluded, is there still a significant difference between TB and controls? Otherwise you are evaluating the effects of a 2nd intervention based on a positive margin in stead of TB staining. Please comment.
- please explain how TB staining will guide the surgeon during resection of the deep margins, while TB has been applied to the superficial mucosa/tumor?
- is using TB itself a predictor for no development of a local recurrence? Is Chi square significant comparing TB yes/no with recurrence yes/no?
- What is your definition of a local recurrence and recurrence free survival
- 193-194:the authors state that the better survival in the TB groups is a consequence of the negative margins obtained in the TB group and suggest that TB is contributing to a better survival. However, in line 143 they state that positive margins is the ONLY predictor for local recurrence. Please comment.
- did the 5 local recurrences develop at the same location as the original primary tumor, or should they be considered as new primary tumors (especially those who arise more than 2 years after surgery)? Please comment.
- Figure 3: please explain why this difference is statistically significant despite crossing lines ?
Minor remarks:
- abbreviations:
- TLM is used for Transoral Laser Surgery , do the authors mean TLS or Transoral Laser Microsurgery ?
- OHPL (51) is not explained before
- RMN (81) : MRI?
- 79: as a control group
- 81: informed consent: do the authors mean informed consent for this study (which is hard to belief since it is an historic group) or for the surgery? Please specify.
- 83: was CT na/or MRI performed in all patients / equally distributed between TB and control group? If not, could this introduce bias? if so, please refer to in Discussion.
- 96: clinical database will do.
- does TB staining impair histopathological analysis?
- In Figure 3, no censored cases are shown. please adjust the figure by showing these censored cases.
Author Response
ANSWERS TO THE REVIEWER 2
75: in cases of T1 laryngeal carcinomas: could previous biopsies influence TB staining?
As has been demonstrated in previous publications reported in the introduction and in the discussion, TB staining is able to discriminate the inflammatory mucosa from dysplastic or carcinomatous mucosa, therefore a previous biopsy has no influence on the reliability of the results.
77-78: 23 TB and 21 control patients in 3-4 years i.e. six/seven T1-2 laryngeal cancer patients each year on average: are these all T1-2 laryngeal cancer patients treated in your institute during this period? Are they consecutive patients or has there been a selection (bias)? This should be explained.
The patients taken into consideration were all treated at the same Institution (University of Catanzaro) and always by the same first surgeon (Prof. Aldo Garozzo, Line 81) and were selected according to the inclusion and exclusion criteria listed in materials and methods.
124-125: Is it statistically sound to evaluate recurrence free survival after a second intervention ? If these patients are excluded, is there still a significant difference between TB and controls? Otherwise you are evaluating the effects of a 2nd intervention based on a positive margin in stead of TB staining. Please comment.
Patients undergoing re-intervention for positive margins were not considered among the patients in whom the recurrence appeared. They should be considered with persistent disease after the first treatment , then the re-intervention has been performed in order to obtain the oncological radicality. In all these patients resection margins were negative and were not subjected to any further treatment. We added these considerations (lines 131-133).
please explain how TB staining will guide the surgeon during resection of the deep margins, while TB has been applied to the superficial mucosa/tumor?
The TB ability to penetrate into the deep layers of the mucosa is not known and the study has shown that TB is reliable in the evaluation of the superficial margins (P = 0.047), while not proven useful in the deep margin assessment (Lines 148-149 and table 2)
is using TB itself a predictor for no development of a local recurrence? Is Chi square significant comparing TB yes/no with recurrence yes/no?
By analyzing the number of patients who developed or not a recurrence, in the TB group 25% of patients with positive close / positive margins developed a relapse while none of the negative patients developed a relapse. This difference was not statistically significant, probably due to the limited number of cases and the fact that the positives were immediately submitted to revision of the surgery (lines 145-146).
What is your definition of a local recurrence and recurrence free survival
The auditor's question seems somewhat strange to me. My definition corresponds to the internationally shared definition. I have over 30 years' experience in the field of head-neck oncology, so I assure the Reviewer that I know these definitions.
193-194:the authors state that the better survival in the TB groups is a consequence of the negative margins obtained in the TB group and suggest that TB is contributing to a better survival. However, in line 143 they state that positive margins is the ONLY predictor for local recurrence. Please comment.
We confirm that considering the total number of patients (n.44) the positive resection margins represented a negative prognostic factor of local recurrence (we specify in the test line 154)
did the 5 local recurrences develop at the same location as the original primary tumor, or should they be considered as new primary tumors (especially those who arise more than 2 years after surgery)? Please comment.
On lines 132-136 we reported in detail the type of recurrence and the type of treatment carried out. Regarding the consideration of whether a relapse that arose after two years can be considered as such or should be considered as a primitive one, I believe that as it is known in the onset and progression of aerodigestive pathway tumors a decisive role has been attributed to cancer stem cells . Cell clones consisting of cancer stem cells can remain quiescent for a long time, moreover their ability to migrate has been demonstrated, therefore able to give rise to relapses in the same location or at a distance.
Figure 2:please explain why this difference is statistically significant despite crossing lines ?please explain why this difference is statistically significant despite crossing lines ?
I thank the Reviewer for detecting this error. We corrected the error in the text and modified Figure 3 by adding the censored ones and comment in the discussion (lines 242-244).
abbreviations:
TLM is used for Transoral Laser Surgery , do the authors mean TLS or Transoral Laser Microsurgery ? Transoral Laser Microsurgery (we specify in the text)
OHPL (51) is not explained before : Thank you we add in the text the definition line 51
RMN (81) : MRI?
Yes, we correct
79: as a control group.
Thanks, we correct
81: informed consent: do the authors mean informed consent for this study (which is hard to belief since it is an historic group) or for the surgery?
Please specify. We specify in the text line 83
83: was CT na/or MRI performed in all patients / equally distributed between TB and control group? If not, could this introduce bias? if so, please refer to in Discussion.
We performed a CT / or MRI according to the same criteria, described in the materials and methods, both in TB patients and in the control group (Line 85).
96: clinical database will do.
I didn't understand what the Reviewer means
does TB staining impair histopathological analysis?
As described in the introduction and discussion TB is a vital dye, it does not alter cell morphology
In Figure 3, no censored cases are shown. please adjust the figure by showing these censored cases.
We replied above.
Round 2
Reviewer 1 Report
Thank you for your resubmission.
The revised manuscript has some improvements with most concerns addressed.
In the discussion, the concluding sentence should be revised. It is comparing to something (NBI?) but the data had no actual comparisons to draw these conclusions.
Potential revision:
'We believe that the use of intraoperative toluidine blue staining has some advantages since it is inexpensive, simple and rapid that is minimally technique-sensitive method.'
In addition, the conclusions do not address the fact that a larger cohort would be beneficial for power, but the current study methodology would benefit from a more valid (i.e. not historical) control group.
Author Response
The revised manuscript has some improvements with most concerns addressed.
In the discussion, the concluding sentence should be revised. It is comparing to something (NBI?) but the data had no actual comparisons to draw these conclusions.
Potential revision:
'We believe that the use of intraoperative toluidine blue staining has some advantages since it is inexpensive, simple and rapid that is minimally technique-sensitive method.'
In addition, the conclusions do not address the fact that a larger cohort would be beneficial for power, but the current study methodology would benefit from a more valid (i.e. not historical) control group.
I'm sorry, but unfortunately I didn't understand the Reviewer's suggestions well. However, I believe that in the discussion and in the conclusions we have clarified the critical issues due to the number of patients and the type of control group and we have also reported the advantages of the method.
Reviewer 2 Report
Thank you for resubmission.
Please define local recurrence and recurrence free survival in methods section. I do not doubt the level op experience, but in many papers regarding transoral laser surgery different definitions of a locla recurrence or recurrence free survival are used. Some studies report a new tumor 5 years after treatment as a local recurrence, which is not right. Other authors report time between the first excision and time of recurrence as recurrence free survival, while other use the second excision (even if this is carried out after 4 months). Nowadays in Journals like J Clin Onc these definitions are mentioned in the Methods section. Please define local recurrence and recurrence free survival, just to compare your results with other studies.
other questiions are answered sufficiently: thank you very much!
And good luck with this interesting approach!
Author Response
Please define local recurrence and recurrence free survival in methods section.
Many thanks to the Reviewer, in the materials and methods we have added the required definitions. Lines 97-100